# Identification of Indoor Air Quality Factors in Slovenian Schools: National Cross-Sectional Study

**An Galičič [1,2,*], Jan Rožanec [1], Andreja Kukec [1,2], Tanja Carli [1,2], Sašo Medved [2,3] and Ivan Eržen [1,2]**

1   National Institute of Public Health, Trubarjeva ulica 2, SI-1000 Ljubljana, Slovenia
2   Faculty of Medicine, University of Ljubljana, Vrazov trg 2, SI-1000 Ljubljana, Slovenia
3   Faculty of Mechanical Engineering, University of Ljubljana, Aškerčeva cesta 6, SI-1000 Ljubljana, Slovenia
*   Correspondence: an.galicic@nijz.si

**Abstract:** Poor indoor air quality (IAQ) in schools is associated with impacts on pupils' health and learning performance. We aimed to identify the factors that affect IAQ in primary schools. The following objectives were set: (a) to develop a questionnaire to assess the prevalence of factors in primary schools, (b) to conduct content validity of the questionnaire, and (c) to assess the prevalence of factors that affect the IAQ in Slovenian primary schools. Based on the systematic literature review, we developed a new questionnaire to identify factors that affect the IAQ in primary schools and conducted its validation. The questionnaires were sent to all 454 Slovenian primary schools; the response rate was 78.19%. The results show that the most important outdoor factors were the school's micro location and the distance from potential sources of pollution, particularly traffic. Among the indoor factors, we did not detect a pronounced dominating factor. Our study shows that the spatial location of schools is key to addressing the problem of IAQ in schools.

**Keywords:** primary school; indoor air quality factors; outdoor air quality factors; questionnaire; cross-sectional study





## 1. Introduction

Research shows [1] that people spend around 90% of their time in indoor environments (housing, public buildings, educational settings, etc.). Data show that school-aged children and adolescents spend almost 12% of their lives in the school environment, significantly more time than in any other indoor environment except the indoor living environment at home [2]. Indoor air is known to have equal or greater impacts on health than outdoor air [3]. Indoor air quality (IAQ) in schools is recognized as one of the most important risk factors affecting pupils' health and learning performance [2]. Children and adolescents are more susceptible to the effects of air pollution than adults, because they typically have a higher respiratory rate due to a faster metabolism, which means higher exposure to pollutants in relation to body weight compared to adults [4]. Their lungs and immune systems are still not fully developed, which makes children more prone to frequent respiratory infections. Therefore, children and adolescents, aged 13 years or younger, are classified as a vulnerable population group [5].

Several international studies have been conducted in Europe on the quality of the school environment and its impact on children's health. Simoni et al. [6] reported that in observed schools (Italy, France, Norway, Sweden, and Denmark), the mean concentrations of $CO_2$ exceeded 1000 ppm in 66% of the included classrooms and the mean concentrations of $PM_{10}$ were elevated over 50 µg·m$^{-3}$ in 78% of the classrooms. Respiratory symptoms were more frequently self-reported and parent-reported for children from poorly ventilated classrooms. Szabados et al. [7] have measured concentrations of $PM_{2.5}$ above the World Health Organization recommended levels in 85% of schools (Czech Republic, Hungary, Italy, Poland, and Slovenia). About 80% of schools had concentrations of $CO_2$ above

1000 ppm. For 31% of school buildings, it was found that exposure to indoor air pollutants could present a significant health risk. In addition, the median lifetime cancer risk value exceeded the acceptable value for radon and formaldehyde.

From a public health perspective, there is a need to improve the IAQ in schools. As IAQ in schools is not always monitored, it is important to consider the identified factors on IAQ at all stages of the design, construction, and management of school buildings [7]. To improve the indoor environment quality of buildings, it is important to obtain data directly from the users. At the international level, the post-occupancy evaluation methodology is one of the methods used to identify the factors that affect the indoor environment [8]. The benefits of such an assessment include obtaining feedback from users about problems in buildings and in identifying solutions; the feed-forward of the positive and negative lessons learned into the next building cycle; and the creation of databases and designing protocols [9]. In this way, a post-occupancy evaluation can be used to support technical measures to improve the performance of indoor environments. The most common way of obtaining information using the post-occupancy evaluation is through the use of questionnaires [8].

Our research aimed to identify the factors that affect the IAQ in primary schools, so we set the following objectives: (a) to develop a questionnaire to assess the prevalence of factors in primary schools, (b) to conduct content validity of the questionnaire, and (c) to assess the prevalence of factors that affect the IAQ in Slovenian primary schools. The contribution to the related literature that this article intends to offer is a new approach to identifying factors that affect the IAQ in primary schools and the example of its application in Slovenian primary schools. In addition, in the discussion, the article explains how identified factors may affect IAQ in schools, thereby contributing to a better understanding of the indoor environment. Last but not least, the article suggests possible measures that could be taken to improve the quality of the school's indoor environment to protect children's health in the future.

## 2. Materials and Methods

### 2.1. Identification of the Factors That Affect the Indoor Air Quality and Development of a Questionnaire to Identify These Factors

The development process was performed in two phases. In the first phase, we identified the factors that affect the IAQ, followed by the development of the questionnaire.

First phase: the identification of the factors that affect the IAQ was based on a systematic literature review in the ScienceDirect database. The purpose of the literature review was to identify the factors that affect the IAQ in the school environment. The search term used was "IAQ" OR "IAP" OR "indoor air" AND "school" OR "classroom" OR "kindergarten" OR "primary education" AND "risk factors" OR "environmental factors" for the period from 2010 to 2019. The literature was selected based on inclusion and exclusion criteria, which were designed according to the purpose of the literature review. The screening of the results of the selected search term was completed in five steps and according to the Preferred Reporting Items for Systematic review and Meta-Analysis (PRISMA) [10]. The inclusion and exclusion criteria for each step are shown in the online Supplementary Materials, Table S1. Of the 514,905 studies in first step of the systematic review, we included 72 that were relevant. More detailed results are shown in the online Supplementary Materials, Figure S1. The most frequently identified sources of indoor air pollution were the proximity to busy roads ($n = 57$) and the classroom activity of the users ($n = 46$). In addition to traffic, the researchers also identified the following outdoor factors that affect the IAQ: commercial and industrial establishments ($n = 29$); meteorological conditions ($n = 25$); emissions from heating buildings ($n = 20$); compounds from the natural environment ($n = 17$); atmospheric reactions and secondary emissions ($n = 11$); unpaved school playgrounds ($n = 6$); and smoking ($n = 3$). For the indoor factors that affect the IAQ, besides the classroom activity, they also identified: ventilation ($n = 35$); cleaning processes ($n = 33$); age and number of children/occupation rate ($n = 30$); technical characteristics of the classroom/building

(*n* = 29); and materials and equipment (*n* = 28). See the online Supplementary Materials, Table S2.

Second phase: The questionnaire has been prepared based on the identified factors that affect the IAQ. The questionnaire included the following three sections: (1) school building and school location information, (2) 3rd grade classroom information and IAQ in the classroom, and (3) natural ventilation of the classroom (Table 1).

**Table 1.** Content of the developed questionnaire in Slovenian primary schools.

| **Section 1: School Building and School Location Information** | **Section 2: 3rd Grade Classroom Information and Indoor Air Quality in the Classroom** | **Section 3: Natural Ventilation of the Classroom** |
|---|---|---|
| Location (statistical region, micro location); building (year of construction, year of last extension or renovation, purpose of construction); sources of outdoor air pollution (potential sources within 200 m, proximity to a major road); heating (type of heating, period of the heating season). | Classroom (year of construction, year of the last renovation, number of pupils during class time, type of flooring, the height of the ceiling, window surface, orientation); materials (flooring, window frames); equipment (type of board and writing equipment, humidifiers, air fresheners); cleaning (frequency, method, cleaning schedule, ventilation during cleaning); classroom damage (damp spots, mold growth); classroom activity (school breakfast); perceived air quality (according to season and heating season). | Ventilation frequency (heating season, non-heating season); ventilation efficiency (ventilation duration, window opening method); human factor (giving incentive for the ventilation by children, the reason for less ventilation by opening windows). |

### 2.2. Content Validation of the Questionnaire

The questionnaire was validated in terms of its content validity [11] and face validity [12]. Content validation was performed among 6 experts (4 public health experts, 1 expert in the field of ventilation, and 1 in the field of school infrastructure) who were asked to give a score of either 0 (item not relevant) or 1 (item very relevant). Of the Content Validity Indices (CVIs), we calculated: the scale content validity index (S-CVI/Ave), scale universal agreement validity index (S-CVI/UA), and a face item validity index (I-FVI). This was followed by response process validation among 12 raters who were asked to give a score of 0 or 1 based on the clarity and comprehensibility of the questionnaire. For the indices, the following threshold was set: I-CVI ≥ 0.78 [11] and S-CVI/Ave ≥ 0.90 [13], and S-CVI/UA a value of ≥0.80 [14,15] and I-FVI above ≥0.83 [12].

The questionnaire was clear to all participants. On average, it took 20 min to answer all the questions. The final version of the questionnaire included 38 questions. The I-CVI and S-CVI/Ave reached a value of 1.00 and the S-CVI/UA reached a value of 0.97, while the I-FVI reached a value of 0.85.

Therefore, during the content validation of the questionnaire, we gave the whole questionnaire to 12 3rd grade teachers to complete. We checked with them their understanding of the whole questionnaire and the correctness of the answers. In accordance with their minor comments, we upgraded the questionnaire to make it fully understandable for the teachers.

### 2.3. Assessment of the Prevalence of Factors That Affect the Indoor Air Quality

The national cross-sectional study on IAQ and natural ventilation of classrooms in Slovenian primary schools (3rd grade) was carried out between 7 January 2020 and 6 February 2020. The population surveyed included all 454 Slovenian primary schools in the school year 2019/2020. The observation unit was the 3rd grade classroom of each primary school. We selected the 3rd grade because pupils in the 3rd grade in Slovenia are in the same classroom for the entire duration of classes.

The request for participation, a participation/informed consent form, and study questionnaires were sent to all primary schools by traditional mail. In the informed consent form, they agreed that they were aware of the purpose and meaning of the study and

that they were willing to participate in it. They had the opportunity to ask questions or ask for help in completing the questionnaire via the researcher's email address. The questionnaire was addressed to the headmasters of the primary schools who selected the 3rd grade teachers who filled out the questionnaire with the assistance of the caretaker. The questionnaires were sent to the primary schools on 7 January 2020 and the collection was completed when the last questionnaire was received on 6 February 2020.

The response rate was 78.19%, which represents a response from 355 out of 454 primary schools in Slovenia.

The distribution of values of the technical characteristics of classrooms is shown by the statistical parameters minimum and maximum, quartile 1, median, quartile 3, average, and standard deviation. With a univariate statistical analysis, we assessed the association between the outdoor factors of IAQ and the micro location in Slovenian primary schools, the association between the indoor factors of IAQ and the year of construction, and the association between the outdoor factors (classroom discomfort, IAQ, and outdoor noise) and the micro location. A univariate statistical analysis was performed using the Pearson, Chi-Square, or Fisher's exact test. The statistical significance was defined at $p \leq 0.05$. Data analyses were made in SPSS (version 27).

The research was approved by the Medical Ethics Committee of the Republic of Slovenia (No. 0120-548/2019/4).

## 3. Results

### 3.1. School Building and School Location Information

The data from the first section on the characteristics of the school building and school location show that most schools are located in villages/rural areas, of which most were built between 1960 and 1979 and mainly expanded/renovated in the period from 2010 to 2019. More detailed information on school buildings and school location is shown in Table 2, and the technical characteristics of the 3rd grade classrooms are shown in Table 3.

**Table 2.** School building and school location information in Slovenian primary schools.

| Variables | Description of Variables | Number | Prevalence [%] |
|---|---|---|---|
| Primary school micro location ($n$ = 334) | City center | 49 | 14.67 |
| | Suburbs/small town | 124 | 37.13 |
| | Village/rural area | 161 | 48.20 |
| Year of school construction ($n$ = 320) | Until 1959 | 98 | 30.63 |
| | 1960–1979 | 147 | 45.94 |
| | 1980–1999 | 50 | 15.63 |
| | 2000–2019 | 25 | 7.81 |
| Year of extension and/or last renovation ($n$ = 286) | Until 1959 | 1 | 0.35 |
| | 1960–1979 | 9 | 3.15 |
| | 1980–1999 | 40 | 14.00 |
| | 2000–2019 | 236 | 82.52 |
| Was the school building built for the purpose of education? ($n$ = 350) | Yes | 347 | 99.14 |
| | No | 3 | 0.86 |
| The floor where the 3rd grade classroom is located ($n$ = 353) | Ground floor | 151 | 42.78 |
| | 1st floor | 157 | 44.48 |
| | 2nd floor | 39 | 11.05 |
| | 3rd floor | 5 | 1.42 |
| | Mansard | 1 | 0.28 |

**Table 3.** Technical characteristics of the 3rd grade classrooms in the Slovenian primary schools.

| Technical Characteristic | Min | Q1 | Average | Q3 | Max | SD |
|---|---|---|---|---|---|---|
| Flooring surface [m$^2$] | 20.00 | 50.55 | 59.70 | 62.50 | 96.00 | 10.71 |
| Ceiling height [m] | 2.10 | 3.00 | 3.30 | 3.60 | 5.50 | 0.52 |
| Window surface * [m$^2$] | 1.00 | 8.88 | 12.00 | 16.00 | 40.00 | 6.39 |

* Total ventilation area of all windows in the classroom; Min—minimum, Q1—first quartile, Q3—third quartile, Max—maximum, SD—standard deviation.

### 3.2. The Outdoor Factors of Indoor Air Quality

The results of the outdoor IAQ factors show that within 200 m from the school, the most frequent potential source is a busy road and a residential area with individual wood-burning stoves. Over half of the schools (56.43%) are located within 100 m of a busy road. The prevalence of outdoor IAQ factors and the association between them and the micro location is shown in Table 4.

### 3.3. The Indoor Factors of Indoor Air Quality

The results of the indoor IAQ factors show data on the materials used in the classrooms, cleaning characteristics, the occurrence of moisture-related factors, and the location of the school breakfast, either in the classroom or in the dining hall. The prevalence of indoor IAQ factors and the association between them and the year of construction is shown in Table 5.

**Table 4.** Prevalence of outdoor factors of indoor air quality and the association between the outdoor factors of indoor air quality and the micro location in Slovenian primary schools.

| Variables | Description of Variables | City Center: Number (Prevalence [%] for This Micro Location) | Suburbs/Small Town: Number (Prevalence [%] for This Micro Location) | Village/Rural Area: Number (Prevalence [%] for This Micro Location) | Total: Number (Prevalence [%]) | *p* |
|---|---|---|---|---|---|---|
| Are there potential sources of air pollution located within 200 m from the primary school? (*n* = 350) ** | Busy road | 39 (45.35) | 86 (42.16) | 97 (39.59) | 222 (63.42) | 0.030 |
| | Industrial zone | 6 (6.98) | 10 (4.90) | 2 (0.82) | 18 (5.14) | 0.003 |
| | Individual industrial installations | 8 (9.30) | 18 (8.82) | 12 (4.90) | 38 (10.86) | 0.088 |
| | Residential areas with individual wood-burning stoves | 24 (27.91) | 71 (34.80) | 106 (43.27) | 201 (57.43) | 0.076 |
| | No potential sources of pollutants can be identified in the school's surroundings | 8 (9.30) | 14 (6.86) | 21 (8.57) | 43 (12.29) | 0.670 |
| | Other | 1 (1.16) | 5 (2.45) | 7 (2.86) | 13 (4.39) | * |
| Distance from the nearest thoroughfare (not the access road to the primary school) (*n* = 350) | 0–100 m | 28 (65.12) | 60 (53.10) | 81 (56.25) | 169 (48.29) | <0.001 |
| | 101–200 m | 7 (16.28) | 16 (14.16) | 23 (15.97) | 46 (13.14) | <0.001 |
| | 201–500 m | 6 (13.95) | 25 (22.12) | 26 (18.06) | 60 (17.41) | <0.001 |
| | 501–1000 m | 2 (4.65) | 8 (7.08) | 8 (5.56) | 18 (5.14) | <0.001 |
| | >1000 m | 0 (0.00) | 4 (3.54) | 6 (4.17) | 10 (2.86) | * |
| Energy source used to heat the school building (multiple answers) (*n* = 355) ** | Natural gas | 18 (36.73) | 61 (49.19) | 40 (24.84) | 119 (33.52) | <0.001 |
| | Fuel oil | 7 (14.29) | 22 (17.74) | 40 (24.84) | 69 (19.44) | 0.167 |
| | Solar cells | 7 (14.29) | 22 (17.74) | 31 (19.25) | 60 (16.90) | 0.728 |
| | Heat pump | 0 (0.0) | 1 (0.81) | 0 (0.0) | 1 (0.28) | 0.669 |
| | Firewood | 2 (4.08) | 16 (12.90) | 49 (30.43) | 67 (18.87) | <0.001 |
| | Wood pellets, wood chips, wood briquettes | 22 (44.90) | 20 (16.13) | 19 (11.80) | 61 (17.18) | <0.001 |

**Table 4.** *Cont.*

| Variables | Description of Variables | City Center: Number (Prevalence [%] for This Micro Location) | Suburbs/Small Town: Number (Prevalence [%] for This Micro Location) | Village/Rural Area: Number (Prevalence [%] for This Micro Location) | Total: Number (Prevalence [%]) | *p* |
|---|---|---|---|---|---|---|
| Orientation of the classroom (*n* = 353) ** | Towards a traffic road | 7 (14.29) | 18 (14.52) | 25 (15.53) | 50 (14.16) | 0.962 |
| | Towards a road with moderate traffic | 19 (38.78) | 31 (25.0) | 40 (24.84) | 90 (25.50) | 0.130 |
| | Towards the school car park | 5 (10.20) | 24 (19.35) | 26 (16.15) | 55 (15.58) | 0.339 |
| | Towards school playground | 13 (26.53) | 35 (28.23) | 51 (31.68) | 99 (28.05) | 0.717 |
| | Towards school grounds park | 11 (22.45) | 38 (30.65) | 36 (22.36) | 85 (24.08) | 0.246 |
| | Other | 5 (10.20) | 13 (10.48) | 19 (11.80) | 37 (10.48) | * |

* Data not provided; ** There were multiple possible answers to this question.

**Table 5.** Prevalence of indoor factors of indoor air quality and the association between the indoor factors of indoor air quality and the year of construction in Slovenian primary schools.

| Variables | Description of Variables | Number | Prevalence [%] | *p* |
|---|---|---|---|---|
| Flooring material (*n* = 355) ** | Parquet | 151 | 42.54 | <0.001 |
| | Laminate | 6 | 1.69 | 0.674 |
| | Synthetic materials (linoleum panels, vinyl panels, PVC, etc.) | 197 | 55.49 | <0.001 |
| | Other | 2 | 0.56 | * |
| Type of board and writing equipment (*n* = 346) ** | Green chalkboard and chalk | 282 | 81.50 | 0.339 |
| | Plastic whiteboard and markers | 169 | 48.84 | 0.035 |
| | Interactive whiteboard and associated digital pen | 142 | 41.04 | 0.620 |
| Frequency of classroom cleaning (*n* = 353) | Several times a day | 28 | 7.93 | * |
| | Once a day | 324 | 91.78 | * |
| | Every other day | 1 | 0.28 | * |
| Classroom cleaning method (*n* = 346) | Wet cleaning of floors and surfaces | 274 | 79.19 | * |
| | Dry cleaning of floors and surfaces | 43 | 12.43 | * |
| | Combination | 29 | 8.38 | * |
| Classroom cleaning term (*n* = 349) | In the morning before classes | 1 | 0.29 | * |
| | Afternoon after classes | 345 | 98.85 | * |
| | Combination | 3 | 0.86 | * |
| Opening windows during classroom cleaning (*n* = 330) | Yes | 180 | 54.55 | * |
| | Not in winter, yes in summer | 128 | 38.79 | * |
| | No | 22 | 6.67 | * |
| Presence of damp patches on walls, ceiling or floor (*n* = 355) | Yes | 7 | 1.97 | * |
| | No | 348 | 98.03 | * |
| Presence of visible mold growth in the classroom (*n* = 355) | Yes | 2 | 0.56 | * |
| | No | 353 | 99.44 | * |
| Location of the school breakfast (*n* = 351) | In classroom | 208 | 59.26 | 0.508 |
| | In dining hall | 143 | 40.74 | 0.570 |

* Data not provided; ** There were multiple possible answers to this question.

### 3.4. Ventilation of Classrooms

The data from the third section on the natural ventilation of schools shows data on the prevalence and ventilation characteristics in the heating and non-heating seasons, with 30.00% of classrooms being ventilated for more than 45 min in the heating season, while 29.88% of classrooms are ventilated for more than 180 min in the non-heating season (Table 6).

**Table 6.** Prevalence and ventilation characteristics in the heating and non-heating seasons in Slovenian primary schools.

| Variables | Description of Variables | Heating Season | | Non-Heating Season | |
|---|---|---|---|---|---|
| | | Number | Prevalence [%] | Number | Prevalence [%] |
| Ventilation frequency (*n* = 353) ** | Classroom is ventilated before the class | 200 | 56.66 | 198 | 56.09 |
| | Classroom is ventilated during every break | 180 | 50.99 | 144 | 40.79 |
| | Classroom is ventilated during every other break | 60 | 17.00 | 18 | 5.10 |
| | Classroom is ventilated once in the morning | 24 | 6.80 | 13 | 3.68 |
| | Classroom is ventilated during class | 192 | 54.39 | 177 | 50.14 |
| | Classroom is ventilated after school breakfast | 127 | 35.98 | 89 | 25.21 |
| | Classroom is ventilated after the end of the class | 129 | 36.54 | 110 | 31.16 |
| | Classroom is not ventilated | 1 | 0.28 | 2 | 0.57 |
| | Classroom is ventilated continuously or most of the time during class | 54 | 15.30 | 181 | 51.27 |
| Average total ventilation time per day (*n* = 340) ** | Less than 20 min | 124 | 36.47 | 25 | 7.40 |
| | 25–45 min | 114 | 33.53 | 43 | 12.72 |
| | 50–90 min | 66 | 19.41 | 74 | 21.89 |
| | 95–135 min | 19 | 5.59 | 60 | 17.75 |
| | 140–180 min | 9 | 2.65 | 35 | 10.36 |
| | More than 180 min | 8 | 2.35 | 101 | 29.88 |
| How do you mostly open the windows? (*n* = 349) | Opening wide | 170 | 48.71 | 151 | 44.15 |
| | Opening on ventus/horizontally | 148 | 42.41 | 119 | 34.80 |
| | Combination | 31 | 8.88 | 72 | 21.05 |

** There were multiple possible answers to this question.

The initiative to ventilate the classroom is usually given by the teacher (309; 91.15%), the pupils (4; 1.18%), or both (26; 7.67%). The reasons why the teacher chooses to open the windows less frequently than normally would be thermal discomfort in the classroom (cold in heating season, heat in non-heating season) (252; 71.39%); outside noise (100; 29.50%), draughts (86; 25.37%), safety concerns (40; 11.80%), and bad outdoor air quality (33; 9.37%). Among the outdoor factors (classroom discomfort, IAQ, and outdoor noise), only outdoor noise has a statistically significant association with micro location ($p = 0.001$).

## 4. Discussion

In our cross-sectional study in Slovenian primary schools, we found the occurrence of some pollutant sources and factors that affect the IAQ to be statistically significant: the association between the frequency of the factors that affect the IAQ and the micro location of the primary school for a distance of 200 m from the major road and an industrial zone; the distance from the nearest major road for the distances of 0–100 m, 101–200 m, 201–500 m, and 501–1000 m; and the energy source used to heat the school building for wood pellets, wood chips, wood briquettes, firewood, and natural gas. An association between the frequency of the factors that affect the IAQ and the year of construction of the primary

school was found for the classroom flooring materials for parquet and synthetic materials, and the type of board and writing equipment for the plastic whiteboard and markers.

### 4.1. School Building and School Location Information

Almost half of the participating Slovenian primary schools are located in villages/rural areas, which is associated with the dispersed and sparse settlements in Slovenia [16]. Almost half of the buildings in the participating primary schools were built between 1960 and 1979. More than 80% of the participating primary schools have renovated their buildings in the last 20 years. Yang et al. [17] found that newer and renovated buildings compared to older buildings have higher emissions of materials and equipment and are more airtight. However, the older buildings were more prone to outdoor pollution, due to the wear and tear of the materials used, worse installation techniques, and consequently more infiltration of outdoor air into the indoor spaces [18]. Most primary schools in Slovenia (99.14%) were built for education purposes. Most of the observed 3rd grade classrooms included in our study were located in the basements, ground floors, and first floors of the school buildings. Studies show that rooms on the lower floors tend to have higher concentrations of volatile organic compounds (VOC) [19] and radon [20] than rooms on the higher floors. Furthermore, Branco et al. [21] found elevated levels of nitrogen dioxide in ground floor classrooms facing toward the road.

### 4.2. The Outdoor Factors of Indoor Air Quality

Most of the participating schools identified a busy road and a residential area with individual wood-burning stoves as potential sources of air pollution within 200 m from the school. Additionally, in other studies, the most frequently indicated outdoor source of indoor air pollution was proximity to busy roads [22–26]. The results of our study show that the busy roads are statistically significantly associated with the micro locations of the participating primary schools, as are the industrial zones. This association is confirmed by other studies, where higher concentrations of traffic pollutants have been reported in urban kindergartens and schools, compared to kindergartens and schools located in the suburbs or rural areas [24,26–28]. However, the concentrations of traffic pollution in outdoor air are not distributed evenly throughout the urban area. In areas with lower traffic density, lower concentrations of pollutants in outdoor air were measured and therefore better IAQ in classrooms [29]; particle number concentrations also decreased with distance from the city center (the main source of traffic emissions) [22]. Schools located more than 5 km from the city had lower and more stable concentrations of traffic pollutants compared to urban schools [26]. Meanwhile, the highest concentrations of industry pollutants have been recorded in schools around the industrial zones and urban areas [30]. Due to the long-range transport of industry emissions, emissions can also be detected in rural areas or the industry is located in areas close to rural areas [27]. In our study, most of the participating schools report being located within 0–100 m of the nearest major road, followed by 201–500 m, 101–200 m, 501–1000 m, and over 1000 m. Rim et al. [23] also pointed out that many schools are located in close proximity to major roads and located less than 100 m away from them. The results of our study show that the distance to the major road is statistically significantly associated with the micro location of the school. Rim et al. [23] found that schools located closer to main roads had higher concentrations of $PM_{2.5}$, $PM_{10}$, and black carbon compared to schools located further away. The greater distance of schools from the road also had a significant impact on indoor concentrations of carbon monoxide and nitrogen dioxide [31]. The most commonly used energy source for the heating of school buildings in Slovenia is natural gas, followed by firewood, fuel oil, wood pellets, wood chips and wood briquettes, solar panels, and heat pumps. Natural gas and firewood are statistically significantly associated with school micro location. Therefore, in rural areas, Canha et al. [32] found a higher impact of emissions from heating surrounding buildings with wood biomass compared to urban environments. Replacing old wood-burning stoves with modern heating systems that are more ecologically friendly in schools does not show a

measurable improvement in IAQ [33]. From this, we can conclude that the IAQ in primary school classrooms during the heating season depends mainly on the type of heating systems in the area and not so much on the type of energy used to heat the school building.

The largest proportion of Slovenian primary schools had the 3rd grade classroom oriented towards the school playground, followed by the school grounds park and the road with moderate traffic. As many as 14.73% of the 3rd grade classrooms were oriented toward the busy road. The position and orientation of the classroom relative to the outdoor source of the pollutant have a significant impact on changes in IAQ [34]. Reche et al. [19,21] found that traffic pollutant concentrations were higher in classrooms, which were oriented toward the street than in classrooms oriented toward schoolyards. Further, the orientation of the building relative to the playground contributed to the differences in PM concentrations. Amato et al. [35] detected higher concentrations of PM when the room was oriented towards an unpaved playground compared to a paved one. Despite this, PM concentrations were higher in classrooms that were oriented towards the street. Almeida et al. [36] detected higher concentrations of PM in classrooms where the classroom door opened directly onto the playground compared to those where the door opened into the building interior.

In total, 77.42% of Slovenian schools are located within 200 m of a major road. The major road is identified by 67.14% schools as a potential source of IAQ pollution. The traffic-related factors are associated with the micro location of the school. This makes the major road the most important outdoor factor, especially in schools in the city center. From this, we can conclude that, of all the outdoor factors, pupils and teachers in urban schools are the most exposed to pollutants generated by traffic. Especially exposed to traffic pollutants are occupants of classrooms that are oriented towards major roads. Pupils and teachers in suburbs/small towns are most likely to be exposed to pollutants from the industrial zones, to which, of course, they are also exposed in the cities. The exposure of pupils and teachers in villages/rural areas is significantly different, as they are most often exposed to pollutants from individual wood-burning stoves. The results of our study and their evaluation suggest that in order to reduce the impact of outdoor factors that affect IAQ, it is necessary to design measures that target the micro location of the school, including the distance from the major road and other potential sources in the school surroundings. When planning the location of new schools, it is necessary to take into account the sufficient distance from the potential sources of pollution and take great care when locating new activities in the areas and proximity of schools.

### 4.3. The Indoor Factors of Indoor Air Quality

The results of our study showed that in Slovenian primary schools, the most commonly used materials for flooring are synthetic materials and parquet. The building materials used, as well as the furniture and equipment in the classrooms, have a significant association with VOC concentrations [30,37,38]. Poulhet et al. [38] found that building materials have higher formaldehyde emissions in classrooms compared to furnishing materials. The building materials have not always been made of the most emissive materials, but due to their high-volume use and coverage of large areas in space, they consequently have a strong impact on concentrations of pollutant emissions. The main source of formaldehyde emissions at all school locations was the classroom ceiling, which contributed on average around 50% of the total indoor formaldehyde emissions. Flooring materials contributed 4–9% of the total formaldehyde emissions. The results of our study also show that the most commonly used board and writing equipment during lessons are green chalkboards and chalk, which have been linked by researchers to PM emissions [34,39]. This is followed by the use of plastic whiteboards and markers, which researchers have linked to total volatile organic compounds (TVOC) emissions [30,37].

The materials of the flooring, synthetic materials, and parquet, as well as the presence of plastic boards and markers, in our study are associated with the age of the school building. For the indoor factors such as building materials (e.g., flooring, walls, and windows) and interior furnishings (e.g., furniture, type of board, and writing instruments),

it is difficult to find a significant association with the building age, as more than 80% of Slovenian primary school buildings have been renovated in the last 20 years. Similar findings were reported by Rivas et al. [18], who noted that although window type is related to the age of the building at installation, the age of the building itself cannot be related to the type of windows, due to frequent renovation activities.

The most commonly used method for cleaning classroom floors and surfaces in Slovenian primary schools is wet cleaning. Less than 21% of primary schools clean the floors of their classrooms with dry cleaning, i.e., by sweeping, vacuuming, and wiping the dust from surfaces with a dry cloth. The classrooms in almost all schools are cleaned once per day in the afternoon after the end of classes. In Slovenia, 54.55% of schools have their windows open during cleaning, while 38.79% of schools ventilate during cleaning only in the summer. This presents a risk as wet cleaning, which is the most common cleaning method in schools, due to the use of cleaning products, results in higher concentrations of TVOCs and are a potential source of polycyclic aromatic hydrocarbons (PAH) [27,40]. Mishra et al. [41] found that the use of cleaning products in schools contributes up to 41% of indoor TVOC concentrations. Wet cleaning can therefore especially affect the concentrations of limonene and p-tolualdehyde, but also the concentrations of some other hydrocarbons [19,30,37,42,43]. In addition, the use of cleaning products can lead to the formation of new particles, such as secondary organic aerosols, which affect PM concentrations [44]. The formation of secondary particles is caused by the reaction between the ozone and terpenes emitted by cleaning products [22]. As a consequence of cleaning with cleaning products, Viana et al. [45] also traced chlorine emissions in indoor dust. In comparison to wet cleaning, dry cleaning resuspends more PM particles into the air, but we should note that dry cleaning does not result in the additional emissions of cleaning products. Cleaning can lead to increases in $PM_{2-10}$ concentrations [44].

The results of our study showed that 1.97% of classrooms have damp patches on the walls, ceilings, or floors and 0.56% of classrooms have visible mold growth. Mainka et al. [46] point out that the presence of mold on walls has a significant impact on the concentration of fungi in indoor air.

The results of our study showed that the majority of the participating schools in Slovenia have their school breakfast in classrooms and not in the dining hall. This is probably due to the lack of space in schools to accommodate the dining hall in the building or the dining hall capacity being too low for the number of pupils attending school breakfasts. When children have their school breakfast in the classroom, food odors are released into the room, and if children are in the classrooms during break time, this has an effect on higher concentrations of carbon dioxide in the classroom [37]. During breaks, there are usually also children playing in the classrooms with lots of moving and running, which also has the effect on the re-suspension of particulate matters (PM) in the air and a consequent increase in PM concentrations [44,47]. During occupied periods, $PM_{10}$ concentrations can be three to five times higher compared to when pupils are not present in the classroom [44].

The results of our study and their evaluation showed that there are a number of indoor factors that affect IAQ in schools, with some of them being controlled by certifications (e.g., materials and equipment used in the classrooms have to comply with the requirements of the certificate). The results of our study and their evaluation show that there is a need to reduce the impact of indoor factors that affect IAQ in schools. It would be necessary to develop measures that are mainly organization-oriented, which can be controlled by the schools themselves (e.g., by organizing school breakfast and lunch in the dining hall, the sufficient ventilation of classrooms, and preventing damp patches).

### 4.4. Ventilation of Classrooms

The results of our study show that during the heating season, classrooms are ventilated for less time compared to the non-heating season. Only 30.00% of classrooms are ventilated for more than 45 min per day during the heating season, while in the non-heating season, more than half of the classrooms are ventilated continuously or most of the time during

the school day. The results of our study also show that the way a classroom is ventilated depends on the ventilation time itself and the outside temperature in relation to the heating and non-heating seasons. For the heating season in Slovenian primary schools, it is typical that the classroom is ventilated for less time but also more intensively by opening the windows wide, a little less on the ventus/horizontally, but still used for the thermal comfort of the classroom users. Compared to the heating season, the prevalence of combined ventilation (ventilation by opening the windows wide and to the ventus/horizontally) increases significantly in the non-heating season, due to the longer ventilation time of the classroom. Other studies have also found an association between the frequency and method of classroom ventilation by season and indoor/outdoor temperature. Kalimeri et al. [37] have recorded two different ventilation patterns. During the heating season, windows were opened for short periods and at a low frequency. The windows were closed during school hours and only partially opened or even closed during breaks, while in the non-heating season the windows were open most of the time. Laiman et al. [48] found that the air exchange in classrooms was 20% higher during the non-heating season as opposed to what was measured during the heating season. Therefore, the frequency of window opening was primarily related to the indoor/outdoor temperature and, consequently, to the thermal comfort of individuals in the classroom. Elbayoumi et al. [49] observed a lower frequency of ventilation by opening windows when the outside temperature was between 28 °C and 32 °C and when outside temperatures were much higher than indoor temperatures. Ventilation of the classroom through opening windows and doors increased when the outside temperature was between 18 °C and 28 °C and when the indoor temperature was significantly higher than the outside temperature. However, when the outside temperature was below 15 °C and the difference between the indoor and outdoor temperature was high, ventilation was adjusted according to the air quality. The studies show that the researchers' findings on the frequency and method of ventilation are consistent with our findings. It is important to emphasize the importance of ventilation during the non-heating season, as a reduced air exchange in the classroom leads to the accumulation of indoor pollutants [31,32]. The results of our study show that teachers have a major role in classroom ventilation. In only 8.85% of the cases of natural ventilation of classrooms, pupils initiated the opening of the windows. The major role of teachers in classroom ventilation was also found by Korsavi et al. [2], who observed that in 16% of cases of natural ventilation, pupils in 3rd–6th grades would open windows on their own. In comparison to our study, a higher proportion of pupils participated in the ventilation of classrooms, but we should be aware that the results of our study are encouraging, as 3rd grade pupils in Slovenian schools have already suggested the ventilation initiative. The most common reasons given by Slovenian teachers for less ventilation in classrooms were thermal discomfort and outside noises. At the same time, it is important to understand that teachers have a higher comfort temperature compared to pupils and therefore ventilation occurs later [2]. This is particularly evident during the heating season, where lower outdoor temperatures and less frequent ventilation increase carbon dioxide concentrations in the classroom [2,31]. Madureira et al. [50] also reported noise problems as a reason for the lower frequency of opening windows in schools. In Slovenian primary schools, we found that external noise is statistically significantly associated with the micro location of the school.

High classroom occupancy, low classroom volume, and inadequate ventilation during classes can lead to excessive levels of carbon dioxide in classrooms [51]. Several researchers have reported average $CO_2$ concentrations exceeding the recommended carbon dioxide level of 1000 ppm [52] in educational settings in England [2], Portugal [3,50,53], Poland [46], and France [44]. It is important to bear in mind that natural ventilation by opening windows depends on the good ventilation habits of the occupants [2]. Ventilating rooms with mechanical ventilation can help to improve IAQ. Schools with central mechanical ventilation allow for continuous ventilation through mechanical ventilation units and ventilation independent of occupants' good habits [51]. Moreover, natural ventilation by opening windows is not always an appropriate ventilation strategy in kindergartens

and schools located in polluted environments or close to significant outdoor sources of pollutants [23]. Majd et al. [31] found that the number of opened windows in classrooms was significantly associated with the concentration of traffic emissions in the classrooms. Each opened window contributed to an 8.2% increase in the average daily concentration of carbon monoxide in the room. Rim et al. [23] observed that the natural ventilation of kindergarten rooms reduced carbon dioxide concentrations but increased indoor air concentrations of black carbon and particulate matter. Some other authors have also found that ventilation increases the concentrations of pollutants in the indoor environment. Elbayoumi et al. [49] observed a positive association between ventilation and indoor carbon monoxide concentrations. Zhang et al. [54] linked $PM_{2.5}$ and nitrogen dioxide emissions to ventilation and air infiltration in classrooms. Rim et al. [23] observed an association between high indoor concentrations of ultrafine particles, particle number, and black carbon indoors and the ventilation of classrooms (in schools located near busy roads).

The results of our study show different approaches to the natural ventilation of classrooms, where teachers have a major role in classroom ventilation. To improve the effectiveness of natural ventilation in terms of IAQ in the classroom, it is necessary to design a natural ventilation strategy that takes into account the heating/non-heating season. This strategy should also take into account the characteristics of the school micro location with identified outdoor sources of IAQ pollution (e.g., rush traffic hours in the urban area), the most efficient ventilation methods, and the ventilation frequency. The implementation of such a strategy would limit the infiltration of outdoor pollutants into the classrooms through open windows and reduce the concentrations of indoor pollutants from classroom indoor air. Such a strategy would also have an impact on reducing the spread of common respiratory viruses (including SARS-CoV-2) indoors, where pupils and teachers are a potential source.

### 4.5. Limitations and Strengths of Our Study

The studies for the development of the questionnaire were obtained from the ScienceDirect bibliographic database. Perhaps better-quality data could have been obtained if more databases were included. Nevertheless, we assess that the data collected are of sufficient quality to develop the questionnaire, as researchers in this field have in the past already conducted literature reviews in several databases, identifying a large number of duplicates. To date, we have not found such a systematic approach to identifying the factors that affect the increase in indoor concentrations of pollutants. In addition, no cross-sectional study has been conducted to assess the prevalence of these factors in Slovenian primary schools. Our national study is characterized by a high response rate of 78.19% and a large study population, representing all 454 primary schools in Slovenia in the 2019/2020 school year. Validation of the questionnaire at the national level is also an important advantage. Therefore, a first analysis of the IAQ situation at the level of the factors that affect the IAQ was prepared. Our study results will contribute to the approach of improving IAQ in schools at the level of factors that affect the IAQ, since in schools often only the risk factors are analyzed (i.e., pollutants). Knowing the factors that affect the IAQ gives us important insights into the discussed issues and gives us focus on the most pressing questions about reducing the concentrations of pollutants in school classrooms. As a part of the study, we have developed a questionnaire that has been validated, giving it scientific weight. The questionnaire may also be used in other countries, with certain adaptations (e.g., season characteristics).

### 5. Conclusions

The main findings of our national cross-sectional study in Slovenia showed that among the outdoor factors, the most important were the micro location of the school and the distance from potential sources of pollution, particularly the main roads. Among the indoor factors, we did not detect a pronounced dominating factor. This is probably because the indoor environment is equipped with elements that meet the quality standards and

safety ratings. Due to the numerous renovations over the last 20 years, the age of the building does not have a major impact on IAQ. This suggests that in Slovenia, the spatial location of schools is key to addressing the problem of the IAQ in schools. This means that school location planning needs to take into account the sufficient distance from potential sources of pollution and take great care when locating new activities in the areas and proximity of schools. To improve the effectiveness of ventilation in classrooms, where natural ventilation is used, it is necessary to design a natural ventilation strategy that takes into account the characteristics of the school micro location (identified outdoor sources of IAQ pollution), the most effective ventilation methods, the ventilation frequency, and the heating/non-heating season. Where a natural ventilation strategy would not be able to provide adequate air quality in the classroom, it is reasonable to include mechanical ventilation, which also requires proper design, use, and maintenance.

**Supplementary Materials:** The following supporting information can be downloaded at: https://www.mdpi.com/article/10.3390/pr11030841/s1, Table S1: Systematic literature review process; Figure S1: A flow chart of the selection of articles for the systematic literature review; Table S2: The results of the literature review of the factors that affect Indoor Air Quality.

**Author Contributions:** Conceptualization, A.G.; methodology, A.G., J.R. and A.K.; T.C.; statistical analysis, A.G.; writing—original draft preparation, A.G. and J.R.; writing—review and editing, A.K. and T.C.; visualization, A.G. and J.R.; supervision, I.E. and S.M. All authors have read and agreed to the published version of the manuscript.

**Funding:** This article is funded by the project "Measures to manage the spread of COVID-19 with a focus on vulnerable groups of population", which is co-financed by the Republic of Slovenia and the European Union under the European Social Fund in the framework of the EU response to the COVID-19 pandemic (Grant No. C2711-20-054101). The content of this article represents the views of the authors only and is their sole responsibility; it cannot be considered to reflect the views of the European Commission or any other body of the European Union. The European Commission does not accept any responsibility for the use that may be made of the information it contains. The study was conducted also as part of a Slovenian Research Agency (ARRS) project, entitled "Development of the prognostic model of exposure to indoor air pollutants in schools and preparation of evidence-based measures for planning of efficient natural ventilation of the classrooms (No. V3-1904)" and grant No. P3-0429.

**Institutional Review Board Statement:** The research was approved by the Medical Ethics Committee of the Republic of Slovenia (No. 0120-548/2019/4).

**Informed Consent Statement:** Not applicable.

**Data Availability Statement:** Data supporting the findings of this study are available from the corresponding author upon reasonable request.

**Acknowledgments:** We would like to thank all headmasters of the participating primary schools for their high response to the request to participate in the study. We would like to thank the 3rd grade teachers and the caretakers of the participating primary schools for their time and for accurately filling in the questionnaires. We would also like to thank all the colleagues and other members of the project "Development of the prognostic model of exposure to indoor air pollutants in schools and preparation of evidence based measures for planning of efficient natural ventilation of the classrooms (No. V3-1904)" who contributed to the development of our study.

**Conflicts of Interest:** The authors declare no conflict of interest.

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
