# Peer review of "Identification of Indoor Air Quality Factors in Slovenian Schools: National Cross-Sectional Study"

_processes, doi:10.3390/pr11030841_

Round 1

Reviewer 1 Report (Previous Reviewer 2)

The manuscript “Identification of Indoor Air Quality Factors in the Slovenian
Schools: National Cross-Sectional Study" is new version of early article “Assessment of Indoor Air Quality Factors in the Slovenian Schools: National Cross-Sectional Survey". The article shows a statistical description of schools taking into account: construction characteristics, classroom equipment, and the subjective perception of users. The authors introduced many changes that affected the quality of the article.

Author Response

Reviewer 2 Report (Previous Reviewer 4)

Thank you for your consideration of the natural ventilation. I could understand your intention to improve the school environment to protect children’s health. 

Please check again your English grammar.

Author Response

Reviewer 3 Report (New Reviewer)

The paper describes a study to identify the factors that affect IAQ in primary schools in Slovenia.

The topic is interesting and useful, but some suggestions are indicated to improve the quality of the work:

- a literature review should be added, also citing the work Lolli et al., 2022 - Post-Occupancy Evaluation's (POE) Applications for Improving Indoor Environment Quality (IEQ)

- insert a representative scheme of the methodological approach

- the contribution of this article to the available literature and its added value should be better described in the introductory section

Author Response

This manuscript is a resubmission of an earlier submission. The following is a list of the peer review reports and author responses from that submission.

Round 1

Reviewer 1 Report

The importance of this work is high because relates indoor air quality and children's health in schools. Authors provide a tool to assess the factors that affect the indoor air quality. However, author provide information from the survey )in several tables and one figure), but they don't relate this with indoor pollutant concentrations. In my opinion, the research must be supported with indoor air pollutants information instead of being supported solely in the perception of people.

Reviewer 2 Report

The manuscript "Assessment of Indoor Air Quality Factors in the Slovenian Schools: National Cross-Sectional Survey" can be assessed as a statistical description of schools taking into account: construction characteristics, classroom equipment and the subjective perception of users.
The goal of the article is clear to me. In lines between 45 to 48 research questions were formulated, however, no direct answer on these questions was founded at work. In this connection, the following questions are as follows:
1.    What kind of tool to assess the IAQ factors in primary schools was used?
2.    How was the validity of the used tool and what assumptions were made?
3.    What methods were evaluated to assess the prevalence of factors that affect IAQ in primary schools?
4.    Can IAQ be assessed on the basis of subjective assessments, which depend on many parameters?

The authors formulated the following final conclusions.
•    The results of our research and its evaluation show that a number of outdoor and indoor environmental factors are important for IAQ – no specific information is available on how these factors directly affect children's health.
•    To achieve good IAQ, it is also essential to design ventilation systems in schools that will mitigate indoor factors affecting IAQ during winter and summer – no such analyzes were carried out in the work, the conclusion was not supported by its own results.

The following additional comments are as follows:
Line 27 – Due to the small volume of air in indoor environments – what the authors mean, is it the air changes rate?
Line 29 – the concentration of pollutants in indoor air is higher compared to outdoor air – Incorrect thesis, true for not all pollutants
Line 42 – IAQ in schools cannot always be measured – What are the limitations?
Lina 79, Chapter 2.3 – information is missing, they are presented in the results section, rather they should be in this section.
Line 88 – the abstract also mentions the summer period, the surveys were conducted only from 7 January 2020 to 6th February 2020
Line 98 – Chapter 3.1 has the same name as 2.1
Line 111 – Section 3.1.2 should be moved to Section 2.1.2
Line 129, Table 2 – Why are there different numbers of analyzed rooms?
Line 151 – Can the winter survey assess the conditions in other seasons?

Reviewer 3 Report

·       General comments:

-      This work is relevant due to the known impact of the bad indoor air quality conditions in students’ health and school performance.

-      The keywords can be reformulated once some of them are to generalist.

-      The manuscript is not well distributed. The methodology is not well described, because some information was not shared. The results’ discussion is very extensive, and not clearly explained. Some results are not included in the results chapter. It could be better to create subsections, in the results’ discussion chapter, directed to each topic, for a better and clearly reading and understanding. The discussion chapter can be improved. There is not clear information for the Slovenian schools in comparison with the obtained in the literature. The last paragraph of the results section looks like a conclusion.

-      Authors says that the study focus on primary schools. However, the sampling is on 3rd classes. I believe that the activities of a 3rd class are different to the other levels. The data of the literature used to compare the obtained results are not all from primary schools, if I’m not wrong.

-      Along the manuscript are described results in indoor air quality. However, If I understood correctly, there weren’t performed any indoor air quality monitoring studies, so authors do not know the real concentrations of the different pollutants in the Slovenian schools. In this way is not possible to made conclusions of the pollutant concentrations, based on the perception of the participants of the study.

-      In general, the manuscript focuses on an important topic, but some information remains unknown, as the questionnaires that authors says that they performed. How is this questionnaire? What it includes? It is a crucial tool to analyze for a better understand of the manuscript.

Some details:

Line 115: “of 514905” what? It is better to identify what is this number.

Line 117: In the literature review searching, there was not used a filter for primary schools. Am I correct? What were about the founded studies?

Lines 125: the “number of children” is the occupation rate.

Line 131: these tables show the results of the questionnaire. However, in some step of the study, it could be interesting to do a check validation of the information in a small sample to assess, for example, the knowledge and know-how about IAQ of the responsible teacher to fill the questionnaire. I believe that if the person responsible to fill the questionnaire does not have information about indoor air quality, may not answer with quality or can increase the subjectivity of the obtained results.

In this table is also mentioned, at column of section 2, “air quality in the classroom”. My question is how authors assessed this information? It is only based on the perception of the participants, am I correct?

Line 190: The discussion chapter starts with information that is not shared in the results chapter.

Line 194: “some of them”. Which ones?

Lines 291-294: This information was not shown in the results.

Line 339: Authors want to say PM2-10 or PM2.5-10?

Line 393: How they get the information about the student’s ventilation initiative?

Line 407: 55% of which schools?

Line 438: The report of the worst IAQ was based on which methodology? Based on the participants perspective? It is very subjective.

Line 448: Authors should keep the same terminology. Previously was mentioned heating and not heating seasons. Now it is mentioned winter and summer seasons.

Line 457: This paragraph can be a conclusion.

Reviewer 4 Report

This study developed a questionnaire on IAQ in primary schools in Slovenia based on a systematic review and described results obtained from a cross-sectional survey using the questionnaire.

 In general, I find the overall writing, both grammar, and content, to be poor. I would suggest a significant edit be made to correct these errors.

 In the section of “Materials and Methods”, the description of statistical analysis is Insufficient. Please state clearly the relationship authors want to determine. Authors also should describe which software was used to analyze. Does hi2 test mean χ2 test or chi-square?

Please state clearly how the authors obtained informed consent from teachers who responded to the survey.

 Result;

If the authors have the results of the classroom IAQ evaluation by teachers, I recommend analyzing the relationship between IAQ ratings of actual classrooms by them and environmental factors of schools through questionnaires.

 Discussion;

This paper is overall descriptive and lacks in-depth discussions.

Authors should update the discussion part.

Round 2

Reviewer 1 Report

This study is not sufficiently supported with experimental data. Although it is a nationwide study,  there is not enough support to validate the conclusions.

Reviewer 2 Report

I am satisfied with the responses from the authors

Reviewer 4 Report

Thank you for your revision of the manuscript.
I think it has improved and become easier to understand.
However, please recheck the English language. For example, what does it mean ‘exceed’ in line 142?
Basically, I wonder why authors are making suggestions for natural ventilation when the results of this study show that outside air factors have a large impact on IAQ of classrooms of Slovenian primary schools. If a large part of indoor pollution is caused by outside air rather than inside, it is strange to ventilate. Considering the location of schools is important, but it is quite difficult to relocate or rebuild them.
And it is well known that indoor factors also affect IAQ, and ventilation is important to reduce indoor pollutants, so mechanical ventilation, especially ventilation with a filter, is recommended. I think it's one of the suggestions, what do you think?

Author Response

Dear reviewer,

thank you for your review.

The term 'exceed' has been changed to 'assessed'. In addition, we have re-checked the full text of the manuscript.

We agree with your comment, but we would like to highlight that both outdoor and indoor air have an evidence health effect. We understand your comment in the context of other studies that nevertheless conclude that natural ventilation can be effective (Fuoco et al., 2015; Stabile et al., 2016; Stabile et al., 2017). Achieving this effectiveness requires the development of a clear natural ventilation strategy that takes into account all the factors that have been shown to be limiting. In this context, it is also important to consider the microlocation of the school. In the case of the current school buildings, there is a need to make considerations about the ventilation strategy of the school and the location of new buildings in the area and proximity of the school, while when planning a new school building, it is necessary to take into account its location in the space (sufficient distance from potential sources of pollution).

Natural ventilation has its limitations, as does mechanical ventilation. We agree with your proposal, and we would also like to highlight the proper design, use and maintenance of mechanical ventilation. In this way, we have also added this information to our discussion.

Cited studies:

Fuoco FC, Stabile L, Buonanno G, Trassiera CV, Massimo A, Russi A, Mazaheri M, Morawska L, Andrade A. Indoor Air Quality in Naturally Ventilated Italian Classrooms. Atmosphere 2015; 6: 1652–75.

Stabile L, Dell’Isola M, Frattolillo A, Massimo A, Russia A. Effect of natural ventilation and manual airing on indoor air quality in naturally ventilated Italian classrooms. Build Environ 2016; 98: 180–9.

Stabile L, Dell'Isola M, Russi A, Massimo A, Buonanno G. The effect of natural ventilation strategy on indoor air quality in schools. Sci Total Environ 2017; 595: 894–902.